# Identification of a Quality Marker of Vinegar-Processed Curcuma Zedoaria on Oxidative Liver Injury

**DOI:** 10.3390/molecules24112073

**Published:** 2019-05-31

**Authors:** Herong Cui, Beibei Zhang, Guoping Li, Lei Li, Hongshan Chen, Jinchai Qi, Wenxue Liu, Jing Chen, Penglong Wang, Haimin Lei

**Affiliations:** 1School of Chinese Pharmacy, Beijing University of Chinese Medicine, Beijing 102488, China; herongcui@outlook.com (H.C.); 20160241011@bucm.edu.cn (B.Z.); zhenzhu7696@163.com (G.L.); wangyq4515@163.com (L.L.); chs1314as@163.com (H.C.); 17862969559@163.com (J.Q.); 18698765005@163.com (W.L.); 2Tibetan Traditional Medical College, Lhasa 850000, China

**Keywords:** curcuma zedoaria, quality marker, hepatoprotective activity, GC-MS metabolomics, discriminatory components

## Abstract

Curcuma zedoaria (dry stenophora of *Curcuma phaeocaulis* Val., *Curcuma kwangsiensis* S. G. Lee et C. F. Liang, or *Curcuma wenyujin* Y. H. Chen et C.Ling) is a representative herb with clinical effects on liver diseases after being vinegar-processed. The crude Curcuma zedoaria and the processed Curcuma zedoaria (vinegar-boil) have been widely used as mixtures, but their equivalence has not been fully investigated. In this manuscript, quality markers of processed (vinegar-boil) Curcuma zedoaria were investigated by comparison of the compounds and hepatoprotective activities with the crude (three spices) ones. First, GC-MS-based untargeted metabolomics were applied to reveal the discriminatory components and discover potential markers. As a result, a total of six components were identified as potential markers. Then, the hepatoprotective activities were evaluated by dual cell damage models induced by a certain concentration of H2O2 or tertbutyl hydfroperoxide (t-BHP) (55 μM H2O2 or 40 μM t-BHP), which highlighted the potential of the processed Curcuma zedoaria on oxidative stress. Finally, epicurzerenone was identified as its quality marker on oxidative liver injury based on the above results and the cell-based biological assay. Overall, vinegar-processed Curcuma zedoaria was more suitable for the treatment of oxidative liver diseases, and epicurzerenone could be considered as its quality marker.

## 1. Introduction

Traditional Chinese medicine (TCM) is an established effective therapy for complex diseases. However, it lacks sufficient data, especially for clinic guidance and quality control [1,2,3]. Recently, a quality marker was proposed by Liu that refers to the components closely related to the efficacy of TCM. According to the principle, a quality marker must be traceable and closely related to the effectiveness of TCM. This concept has improved the quality standard system with Chinese pharmacopoeia as the key example [4]. Metabolomics have been widely used to investigate the quality markers of TCM, which are able to promote drug discovery and development [5]. The methods of metabolomics can dissect the discriminatory components of TCM more quickly and systematically in order to provide an available guidance for quality control and rational medication [6,7,8].

Curcuma zedoaria (dry stenophora of *Curcuma phaeocaulis* Val., *Curcuma kwangsiensis* S. G. Lee et C. F. Liang, or *Curcuma wenyujin* Y. H. Chen et C.Ling) is a representative herb with clinical effects on liver diseases after being vinegar-processed. Recent pharmacological studies have shown that Curcuma zedoaria has pharmacological effects against cancer, cardiovascular and cerebrovascular diseases, fibrosis, inflammation, bacterial and virus, hypoglycemia, oxidation, dysmenorrhea, and more [9]. In particular, Curcuma zedoaria can be used for the treatment of rectal cancer, gastric cancer, liver cancer, lung cancer, cervical cancer and so on, as well as the effects of anti-platelet aggregation, anti-thrombosis, blood lipid regulation, and anti-atherosclerosis. It also has protective effects on ischemic cerebral apoplexy and anti-liver/kidney/pulmonary fibrosis [10,11]. Recently, it was found that Curcuma zedoaria has obvious hypoglycemic and antioxidant effects with active components including volatile oil, curcumin, curcumol, beta-elemene, and so on [12,13]. For centuries, the crude Curcuma zedoaria of three mentioned species and the processed Curcuma zedoaria (vinegar-boil) have been widely used for modulating the function of the liver and promoting blood circulation [14,15,16]. Components vary with processing conditions. However, these Curcuma zedoarias are used as mixtures because their equivalence has not been fully investigated yet [17]. It is urgent to compare the pharmacological and the chemical changes among the crude and the processed herbs.

In this study, quality markers of Curcuma zedoaria related to the clinical application (oxidative liver injury) were identified by comparison of the crude and the processed ones. Our previous article reported the admirable efficacy of zedoray oil in Curcuma zedoaria in the Liuweiwuling formula on liver failure induced by the lethal dose of D-galactosamine/lipopolysaccharide (GalN/LPS) [18]. This effect might be related to its antioxidant activity. For further cognition in the efficacy of Curcuma zedoaria on liver diseases, we aimed to identify the quality markers of this herb by comparing secondary metabolites and hepatoprotective activities of the crude (three spices) and the processed (vinegar-boil) ones. Specifically, GC-MS-baseduntargeted metabolomics were applied to reveal the discriminatory components of the crude and the processed Curcuma zedoaria. Then, the hepatoprotective activities were evaluated by dual cell models. Finally, the quality markers associated with the hepatoprotective activity were determined based on these results (graphical abstracts). This work provides a reference for the quality control and the clinical guidance of this herb.

## 2. Results

### 2.1. Discriminatory Component Identification

The chemical information of the crude and the processed Curcuma zedoaria was established by GC-MS in the ESI+ mode, and components were tentatively identified by the National Institute of Standards (NIST14) mass spectral libraries. The detailed component identification and fingerprint analysis are depicted in the Appendix A for basic quality control of the objects in this manuscript (Appendix A and Appendix A).

GC-MS-based untargeted metabolomics were applied to reveal the discriminatory components of the crude and the processed Curcuma zedoaria. The base peak intensity chromatograms (BPC) of the samples are presented as Appendix A. The score plots of the principal component analysis (PCA) derived from data of ESI+ mode are shown in Figure 1A. Figure 1B displays the results of the orthogonal partial least-squares discriminant analysis (OPLS-DA) model derived from data of the ESI+ analysis, and the score plot showed good fitness and good predictive ability with a R2Y (cum) of 0.965 and Q2 (cum) of 0.879 of model. All the parameters of the PCA and the OPLS-DA models and the score plots of the PCA analysis that contained quality control (QC) samples are shown in Appendix A and Appendix A. The S-plots (Figure 1C) and variable importance for projection (VIP) values of the OPLS-DA model were used to select the variables responsible for group separation. Variables with a VIP value > 1 and |*p* (corr)| ≥ 0.50 were selected as the variables that were most correlated with the OPLS-DA discriminant scores in order to decrease the risk of false positives in the selection of potential markers. Next, secondary metabolites that differed significantly (*p* < 0.05) in different groups were selected as candidate markers. The criteria were restricted to features with an average normalized intensity difference of 1.5-fold. A total of 10 potential markers and their statistical parameters are detailed in Table 1.

### 2.2. Comparison of Hepatoprotective Activities by L02 Assays

This efficacy of the crude and the processed Curcuma zedoaria on oxidative liver injury was firstly evaluated on normal hepatic cell lines (L02s), and the experimental design is shown in Figure 2A. A certain dose (12.5 μL/mL) of Emodin was set as the positive control in this part of our experiment [19], and 55 μM H2O2 or 40 μM tertbutyl hydfroperoxide (t-BHP) was chosen to be the model conditions according to the previous experiment. MTT assay revealed the inhibiting effects of H2O2 or t-BHP on the viability of L02s by contrast of the control group. This damage was protected by the crude G, W, P group (G: *Curcuma kwangsiensis* S. G. Lee et C. F. Liang; W: *Curcuma wenyujin* Y. H. Chen et C.Ling; P: *Curcuma phaeocaulis* Val.) and the processed C (Curcuma zedoaria) group (83.33 mg/mL) treated with a significant difference (*p* < 0.05) (Figure 2B-D). H2O2 and t-BHP resulted in the damage of L02s within 4 h with an increase of serum aminotransferases and a decrease of total antioxidant capacity (T-AOC). In contrast to the model group, the processed Curcuma zedoaria-treated L02s survived a damaged dose of toxicants, showing normal serum aminotransferase levels and normal T-AOC (Figure 2B–I). Then, we determined the relative fluorescence intensity of reactive oxygen species (ROS) production. Few positive cells were found in the control group. A large number of positive cells was observed at 4 h after toxicants were added. The relative fluorescence intensity was inhibited, showing the potent effect of the processed Curcuma zedoaria on suppressing ROS production induced by different toxicants (Figure 3). Subsequently, the reduction–oxidation (REDOX) balance associated with ROS in L02s was evaluated based on the activity of glutathione (GSH), superoxide dismutase (SOD), and glutathione reductase (GR). Results (Figure 3A-F) showed that the activity of GSH, SOD and GR in the H2O2/t-BHP exposed group was significantly (p < 0.05) lower than in the control group. This indicated that excessive ROS production induced by toxicants could induce oxidative stress, destroy antioxidant defense systems, and promote lipid peroxidation. By contrast, in the C group, the activity of GSH, SOD, and GR in L02s was higher than that in the model group, and the difference was statistically significant (*p* < 0.05). These results indicated that the crude and the processed Curcuma zedoaria could reverse the oxidative damage caused by H2O2/t-BHP. However, abnormal T-AOC was recuperated by the processed Curcuma zedoaria far more than the crude ones.

### 2.3. Comparison of Hepatoprotective Activities by HBMEC Assays

The vascular endothelial cells have an inductive effect on liver development before the establishment of blood flow [20]. Endothelial cells are not only the target cells of free radicals but also the important tissue of free radical production [21]. Stimulating endothelial cells angiogenesis could form the basis of a therapeutic scheme aimed toward protection of hepatocytes, which may have therapeutic potential for preservation of organ function in certain liver disorders [20]. Thus, the efficacy of the crude and the processed Curcuma zedoaria on oxidative liver injury was subsequently evaluated on human brain microvascular endothelial cell (HBMEC) assays, including proliferation, angiogenesis, apoptosis, and ROS production. The cell viability of several concentrations of the crude and the processed Curcuma zedoaria on HBMECs was firstly evaluated by MTT assay, and the results showed that the dose in this manuscript was reliable because of the inhibition of cell viability under 50% (Figure 4A). Then, 55 μM H2O2 or 40 μM t-BHP was chosen to be the model conditions according to the preliminary studies. MTT assay revealed the inhibiting effects of H2O2 or t-BHP on the viability of HBMECs by contrast of the control group. This damage was protected by the crude (G, W, P group) and the processed (C group) Curcuma zedoaria (83.33 mg/mL) treated with a significant difference (*p* < 0.05) (Figure 4B–C). Subsequent studies were only applied with the t-BHP model due to the strong damage of 55 μM H2O2. Furthermore, t-BHP resulted in the damage of HBMECs within 4 h with the blockage of angiogenesis and the increase of apoptosis cells and ROS production with a significant difference (*p* < 0.05) (Figure 4 and Figure 5). In contrast to the model group, the processed Curcuma zedoaria-treated HBMECs survived a damaged dose of toxicants. Specifically, the crude and the processed Curcuma zedoaria could confront the blockage of angiogenesis with a significant difference (*p* < 0.05) (Figure 4D-E, Figure 5A). Then, 4′,6-diamidino-2-phenylindole (DAPI) staining was used for apoptosis detection of cells by morphological analysis. The nucleus of HBMECs in the control group was complete and ovular in shape with weak fluorescence intensity (Figure 5B). The relative fluorescence intensity was inhibited by the crude and the processed Curcuma zedoaria, showing the potent effects of suppressing ROS production (Figure 4E, Figure 5B). These results indicated that the crude and the processed Curcuma zedoaria could reverse the oxidative damage caused by H2O2/t-BHP. However, damaged HBMECs were recuperated by the processed Curcuma zedoaria far more than the crude ones.

### 2.4. Quality Marker Identification

Based on the pharmacological comparison above, the vinegar-processed Curcuma zedoaria was more suitable for the treatment of oxidative liver diseases. Its potential marker identification was conducted using the National Institute of Standards (NIST14) mass spectral libraries. As a result, this quality marker (epicurzerenone) and its integrated peak area in the crude and the processed Curcuma zedoaria are detailed as Table 2. Furthermore, the effects of epicurzerenone on apoptosis in L02 cells induced by H2O2/t-BHP were determined by flow cytometric analysis. The cells were treated by 55 μM H2O2 or 20 μM t-BHP with or without epicurzerenone (32 μM) and then stained with both Annexin V-FITC and PI. The flow cytometry observed four quadrant images: necrotic (Q1; Annexin−/PI+), late apoptotic (Q2; Annexin+/PI+), intact (Q3; Annexin−/PI−), and early apoptotic (Q4; Annexin+ /PI−) cells. The results are shown in Figure 6. The apoptosis ratios (including the early and the late apoptosis ratios) increased to 50.33% (H2O2) and 29.37% (t-BHP), while those of the control and the epicurzerenone groups were respectively 0.98% (control), 14.52% (H2O2+epicurzerenone), and 11.74% (t-BHP+epicurzerenone). Taken together, epicurzerenone was identified as the quality marker of vinegar-processed Curcuma zedoaria associated with the hepatoprotective activity based on these results.

## 3. Discussion

Acute liver failure (ALF) is a rare but life-threatening illness, and medical therapies that interrupt the progression of hepatic injury and the development of extra hepatic organ dysfunction are not readily available [22,23]. Thus, the discovery of complementary and alternative therapies for ALF is extremely urgent. TCM has formed a unique diagnosis and treatment system based on a holistic view, and it is effective for the treatment of complex diseases such as ALF. TCM believes that diseases are caused by inadequate adaptability or overreaction of the body to environmental changes, and local diseases are caused by the imbalance of the whole body; thus, local diseases should be treated by changing the imbalance of the whole body. In a previous manuscript [18], we discovered the effects of zedoray oil in Curcuma zedoaria on rescuing mice from liver failure induced by lethal toxicants via its antioxidant activity. For further cognition in the efficacy of Curcuma zedoaria on liver diseases, we aimed to identify the quality markers of this herb by comparing secondary metabolites and hepatoprotective activities of the crude (three spices) with the processed (vinegar-boil) ones. As a result, with epicurzerenone as one quality marker, vinegar-processed Curcuma zedoaria plays a protective role against hepatocellular and endothelial cell damage caused by oxidative stress induced by toxicants. This work uncovered the antioxidant efficacy of vinegar-processed Curcuma zedoaria on oxidative liver injury in order to guide this herb’s quality control and provide references for clinical medication, mechanisms, and drug discovery of this herb on liver diseases.

Studies have confirmed that main compounds of Curcuma zedoaria (such as curcumin, elemene, and curcumol) have definite therapeutic effects on liver diseases [9,24]. It has been found that curcumin has anti-hepatic injury effects via scavenging free radicals, inhibiting inflammation induced by NF-kappa B, inhibiting the activation of hepatic stellate cells (HSC), and inhibiting oxidative damage induced by ROS [25,26]. Elemene can induce apoptosis of tumor cells, inhibit the growth of tumor cells, and improve immune responses of the body. Curcumol can inhibit hepatic fibrosis by inhibiting the expression of cytokines and inducing apoptosis of tumor cells via inhibiting gene expression [27]. Such literature further confirms the therapeutic potential of Curcuma zedoaria on liver diseases, and its main mechanism is related to anti-oxidative stress and anti-inflammation [24,27,28]. This is consistent with our experimental results. Interestingly, few activity reports of epicurzerenone have been found based on the results of the searching database, which stimulated our interest in further study of this compound.

Processing is one of the characteristics of TCM, and it is the significant difference between TCM and natural medicine. The purpose of processing includes enhancing or changing the potency, changing drug distribution, or reducing drug toxicity in order to facilitate the safe and effective clinical use of TCM. The crude Curcuma zedoaria is used for breaking blood and eliminating accumulation for treating lupus associated nephritis, chronic nephropathy, proteinuria, and chronic renal failure in clinical trials [29]. By contrast, the processed Curcuma zedoaria (vinegar-boil) has been widely used for modulating the function of the liver and promoting blood circulation [30,31]. This manuscript confirmed that the efficacy of the processed Curcuma zedoaria was associated with confrontation against large doses of ROS related oxidative stress. ROS can induce oxidative stress, leading to oxidative damage of DNA, proteins, enzymes, and lipids in cells, resulting in cell death. The enzymatic antioxidant system is one of the antioxidant defense systems in organisms, including SOD, GR, and so on. SOD can catalyze the production of O2-. GR can catalyze the reduction of oxidized glutathione (GSSG) to GSH and has the function of regenerating GSH. When the body is damaged by oxidation, the above antioxidant enzymes will play a great role [32,33]. In this manuscript, abnormal cell activity, transaminase, and total AOC induced by H2O2/t-BHP were recuperated by vinegar-processed Curcuma zedoaria, which was basically due to enhancing the function of the antioxidant defense system and then inhibiting apoptosis of hepatocytes. Further mechanisms will be studied in our next research.

## 4. Materials and Methods

### 4.1. Cell Culture Materials and Chemicals

L02s and HBMECs were provided by the Institute of Peking union medical college. Heat-inactivated fetal bovine serum (FBS) and Dulbecco’s modified eagle medium (DMEM) were obtained from GIBCO Invitrogen (Barcelona, Spain). Penicillin and streptomycin were obtained from Thermo Scientific (Waltham, MA, USA). The 2-Chloro-L-Phenylalanine was obtained from Shyuanye Biochemical Co., Ltd. (Shanghai, China; CAS number 103616-89-3). Methoxyamine hydrochloride (≥98%) and thiazolyl blue tetrazolium bromide (MTT, ≥98%) were obtained from Sigma-Aldrich (St. Louis, Missouri, USA; CAS number 593-56-6). The crude and the processed Curcuma zedoaria and emodin (as the positive control) were purchased from China Pharmaceutical Biological Products Institute, China and were authenticated by Doctor Guijun Zhang (Director, Identification Department of TCM, Beijing University of Traditional Chinese Medicine, Beijing, China). Epicurzerenone (≥98%) was purchased from Daosifu Biological Products Institute, China (CAS number 20085-85-2). Methanol (≥99.9%) and pyridine (≥99%) were obtained from VWR (Leuven, Belgium). Double-distilled water was purified by a Millipore water purification system (Millipore, Bedford, MA). Acetonitrile and methanol (HPLC grade) were purchased from Merck (Darmstadt, Germany) and Burdick & Jackson (Ulsan, Korea), respectively. All other chemicals used were of analytical grade.

### 4.2. Sample Collection and Preparation

Sample collection and preparation were performed using Xiang [31] as reference. The samples were weighed and extracted by ultrasound extraction at 30 kHz for 1 h in a 10 times volume of the mixture solvent (methanol: aether: n-hexane = 1:1:1) that was twelve times the volume of the sample in order to extract as many secondary metabolites as possible from the materials. Each 1.0 mL extract was collected into a glass vial, and 20 μL of 2-chloro-L-phenylalanine (0.3 mg/mL) was added as the internal standard. Subsequently, the solution was mixed in a vortex for 30 s and evaporated to dryness at room temperature under a gentle stream of nitrogen gas. The derivatization process was carried out in two steps. The dried extract was subjected to oximation by adding 50 μL of methoxyamine solution (15 mg/mL methoxyamine hydrochloride in pyridine) and incubated at 70 °C for 60 min, followed by trimethylsilyl (TMS) derivatization by adding 50 μL of BSTFA with 1% TMCS and incubated at room temperature for 60 min. The derivatized residue dissolved in n-hexane was then transferred to autosampler vials after being filtered through a 0.45 μm filter membrane. A 1 μL aliquot of the sample was injected for GC-MS analysis. The seven biological replicates were analyzed with a single technical replicate.

For cell-based biological assays, the crude and the processed Curcuma zedoaria (833.33 mg/mL) were boiled with sterile double distilled water at 100 °C for 4 h based on the clinical dose and processing method and collected by the freeze drying method. Then, samples were dissolved in DMSO and diluted to a final concentration (83.33 mg/mL) in which the content of DMSO was controlled in 0.3% (recognized nontoxic dose).

### 4.3. Gas Chromatography and Mass Spectrometry Settings

The chromatographic analysis of the component fingerprint was performed on an Agilent 7890B/5977 GC-MS system (Agilent Technologies, Palo Alto, CA, USA) using a capillary column 5MS (30mm × 0.25mm × 0.25 μm). Helium C-60 was used as the carrier gas at a constant flow rate of 1.0 mL/min. Samples (2 μL) were injected in split mode (ratio 1:20), and the injector temperature was 250 °C (held for 20 min). The gas chromatography settings were as blows. Total separation run time was 93.174 min. The oven temperature was fixed at 60 °C for 1 min, then increased to 100 °C (rate 4 °C/min) and held for 0 min, then increased to 120 °C (rate 2 °C/min) and held for 0 min, then increased to 180 °C (rate 1 °C/min) and held for 0 min, then increased to 230 °C (rate 23 °C/min) and held for 10 min. The MS detector was operated in ESI mode (70 eV). The ESI temperature was 150 °C. Data acquisition was performed in full scan mode with a mass range between 35 and 450 m/z. Sample preparation and GC-MS acquisition were randomized to avoid analytical bias.

### 4.4. Data Processing and Discriminatory Component Analysis

All data were pre-processed with LineUp (Infometrix Inc., Bothell, WA, USA) and PiroTrans (GL Science Inc. Tokyo, Japan). For molecular feature extraction, peaks with signal-to-noise (S/N) ratios lower than 10 were rejected. The missing value estimation, data filtering, and data normalization were obtained by the MetaboAlalyst 3.0 online software. The resultant data matrices were introduced into the SIMCA-P+ 13.0 (Umetrics, Umeå, Sweden) software for multivariate statistical analysis including PCA and OPLS-DA. Prior to PCA, all variables obtained from the data matrix were mean-centered and scaled to the pareto variance. Variables with a VIP value (VIP ≥ 1.0) and |*p* (corr)| ≥ 0.5 in the OPLS-DA model were selected as potential biomarkers.

Fold changes were calculated by the MetaboAnalyst 3.0 software. The significant differences were further determined by using the fold change value (>1.5) combined with the ANOVA (*p* < 0.05) and t- test (*p* < 0.05). Only the variables with significant changes were selected as potential quality markers and subjected to further identification of their molecular formulas. All markers were tentatively identified by the National Institute of Standards (NIST14) mass spectral libraries (percentage of match of 70% was set as the accepted mass error).

### 4.5. Cell models of Oxidant Stress

H2O2 and t-BHP were respectively used to cause oxidative damage of L02s. L02s or HBMECs were damaged by 55 μM H2O2 or 40 μM t-BHP for 4 h. The crude and the processed Curcuma zedoaria (83.33 mg/mL) was added for 24 h according to clinical doses, respectively. The experiment was repeated three times.

### 4.6. Cell Viability Assays

The hepatocytes or the endothelial cells were seeded in 96-well plates (2.0 × 10^3^ cells/ well) and exposed to the crude and the processed Curcuma zedoaria at a concentration range that included varying concentrations (1.30–83.33 mg/mL) in a 5% CO_2_ containing incubator at 37 °C. The lethal concentration of the hepatocytes or the endothelial cells was estimated according to the MTT reduction assay (with three technical replicates each) and used in subsequent experiments. The procedure was performed as previously described [34]. The optical density for each well was measured on a BIORAD 550 spectrophotometer plate reader at 550 nm.

### 4.7. ALT, AST, Total AOC, GSH, MDA, SOD, and GR Assay

Samples were obtained by ultrasonication for 1 min at 4 °C. The ALT, AST, total AOC, GSH, MDA, SOD, and GR levels were measured according to guides in respective kits (Nanjing Jiancheng, Nanjing, China; CAS number C009-2, C010-2, A015-3, A006-2, A003-1, A001-3, and A062) by a BIORAD 550 spectrophotometer plate reader.

### 4.8. Reactive Oxygen Species (ROS) Assay

The 2′,7′-Dichlorodihydrofluorescein diacetate (DCFH-DA) was used to measure intracellular ROS. DCFH-DA entered the algal cells, reacted with ROS, and then produced the highly fluorescent compound dichlorofluorescein (DCF). Briefly, 1 mL of the algal suspension was collected after centrifugation (9000 g, 5 min) and then washed three times with phosphate buffered saline (PBS). The algal cells were incubated with DCFH-DA (10 μM) in the dark at 25 °C for 30 min. The algal cells were then washed an additional three times with PBS. The fluorescence intensity of DCF was measured using a fluorescence spectrophotometer (ECLIPSE Ts2R-FL, Nikon, Japan) with an excitation wavelength of 488 nm and an emission wavelength of 510 nm.

### 4.9. Tube Formation Assay

Formation of tube networks was assessed as described before (Borradaile and Pickering, 2009; Das, 2018). HBMECs were seeded at 10,000 cells per well in a 24-well plate coated with 10 μL basement membrane matrix (Corning Matrigel, CAS number 356234). The crude or the processed Curcuma zedoaria (83.33 mg/mL) were added to the media. Following a 6 h-incubation, resulting tube networks were analyzed in at least three random fields by light microscopy (@10X, Nikon Eclipse TiE), and the number of branch points and the total length of the tubule networks were quantified by Image-Pro Plus software (version 5.0, National Institutes of Health, USA).

### 4.10. DAPI Staining

The endothelial cells were plated onto 24-well plates at a density of 10,000 cells per well and incubated at 37 °C in a 5% CO_2_ containing incubator. Then, the cells were incubated with or without the crude or the processed Curcuma zedoaria (83.33 mg/mL) for 24 h. Then, cells were fixed with 4% paraformaldehyde (pH 7.4) and dyed by DAPI at the concentration of 1 mg/mL for 5 min in the dark. The cell morphological changes were observed by fluorescent microscopy, and images were captured by a fluorescence spectrophotometer (ECLIPSE Ts2R-FL, Nikon, Japan).

### 4.11. Apoptosis Analysis by Flow Cytometric Using Annexin V-FITC/Propidium Iodide (PI) Staining

The L02 cells were plated onto 6-well sterile plates (1.2 × 10^6^ cells / well) and placed at 37 °C with 5% CO_2_ for 24 h. Then L02s were damaged by 55 μM H2O2 or 20 μM t-BHP for 4 h. Epicurzerenone (32 μM) was added for 24 h. Cells were collected respectively with the right amount of trypsin (without EDTA) digestion, washed with cold PBS, and centrifuged at 1000 rpm for 5 min. The harvested cells were re-suspended in 200 μL binding buffer, which contained Annexin V-FITC and PI, and analyzed with a flow cytometry after the avoided light reaction for 15 min.

### 4.12. Statistical Analysis

The data were analyzed with the SPSS software program (version 22.0, Chicago, IL, USA). One-way ANOVA with the post hoc test followed by the Student’s t-test (the Mann-Whitney U test was used when the t-test was not suitable) was used for the evaluation of significant differences of the results. The differences were considered to be statistically significant when *p* < 0.05 and highly significant when *p* < 0.01. FDR correction was not used during the univariate analysis of the metabolomics analysis, because the metabolites with small *p*-values had been examined by building the classification model [31].

## Figures and Tables

**Figure 1 molecules-24-02073-f001:**
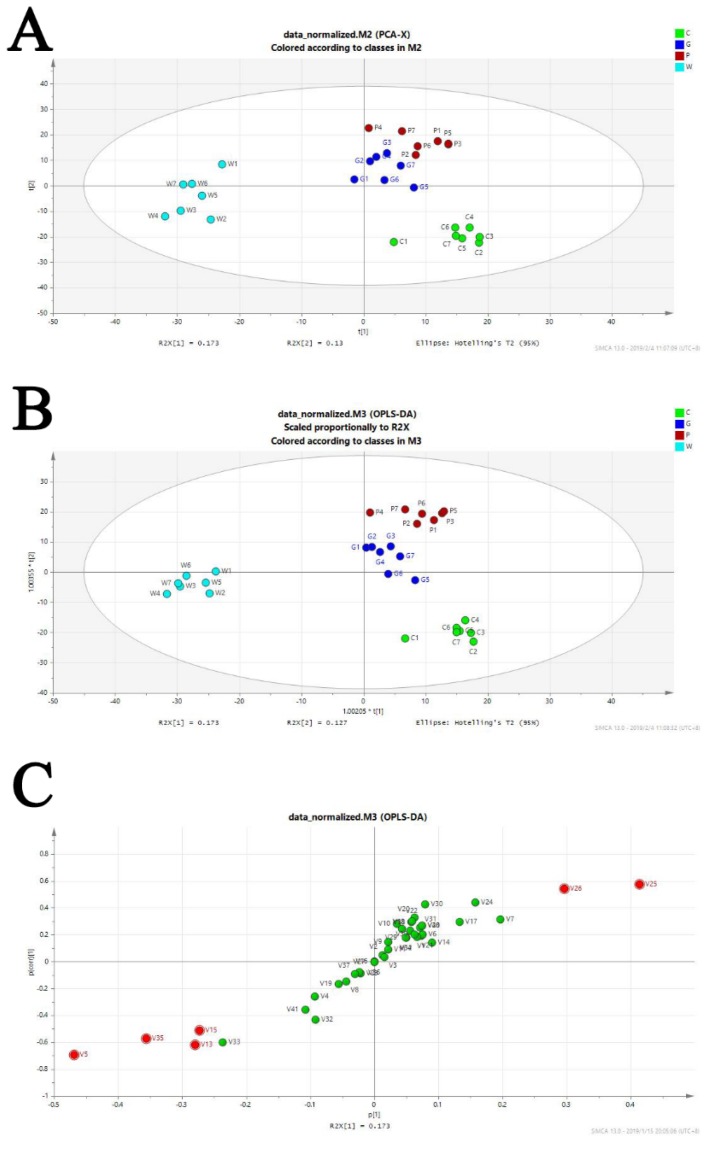
Discriminatory components of the crude and processed Curcuma zedoaria. (**A**) The score plots of the crude (G group for *Curcuma kwangsiensis* S. G. Lee et C. F. Liang; W group for *Curcuma wenyujin* Y. H. Chen et C.Ling; P group for *Curcuma phaeocaulis* Val.) and the processed Curcuma zedoaria control (C group) from principal component analysis (PCA) in the ESI+ mode for PC1 versus PC2. (**B**) Orthogonal partial least-squares discriminant analysis (OPLS-DA) of the data derived from the ESI+ mode. OPLS-DA score plots for the pair-wise comparisons of these four groups. (**C**) S-plot of the OPLS-DA model for these four groups. The points in red indicate the identified markers.

**Figure 2 molecules-24-02073-f002:**
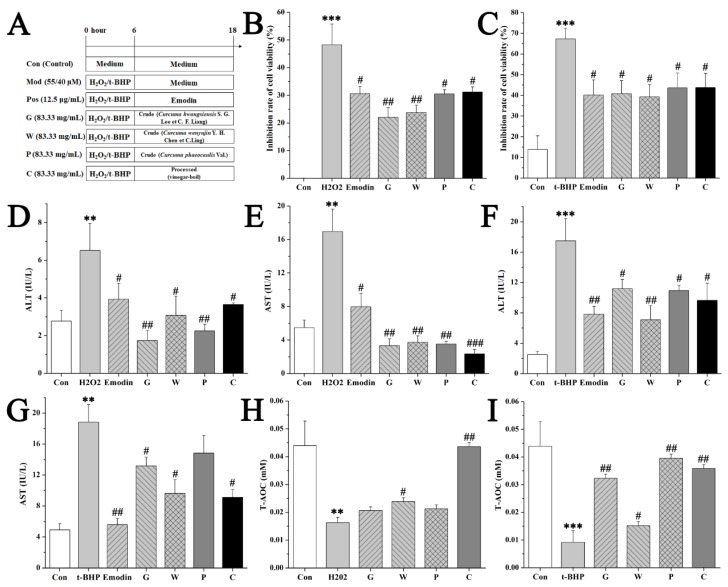
Efficacy evaluation of the crude and processed Curcuma zedoaria on oxidative liver injury by normal hepatic cell line (L02) assays. (**A**) The schematic diagram of the experimental design. Con, control group treated with phosphate buffered saline (PBS); Mod, model group treated with 55 μM H2O2 or 40 μM tertbutyl hydfroperoxide (t-BHP); Pos, positive control group treated with emodin (12.5 μg/mL); G, W, P, C, cells treated with *Curcuma kwangsiensis* S. G. Lee et C. F. Liang, *Curcuma wenyujin* Y. H. Chen et C.Ling, *Curcuma phaeocaulis* Val., and vinegar-processed Curcuma zedoaria at doses of 83.33 mg/mL, respectively. (**B**–**C**) Inhibition rate of cell viability of L02 for MTT assays. (**D**–**G**) The cellular alanine aminotransferase (ALT) and aspartate aminotransferase (AST) levels of H2O2 or t-BHP model, respectively. (H-I) The cellular total antioxidant capacity (T-AOC) levels of H2O2 or t-BHP models, respectively. ANOVA with the post hoc test was used to calculate the significance of the differences, ** and *** represents *p* < 0.01, and *p* < 0.001 compared with the control group; #, ## and ### represent *p* < 0.05, *p* < 0.01, and *p* < 0.001 compared with the model group, respectively. All experiments were performed three times, and the results are expressed as the mean ± S.D.

**Figure 3 molecules-24-02073-f003:**
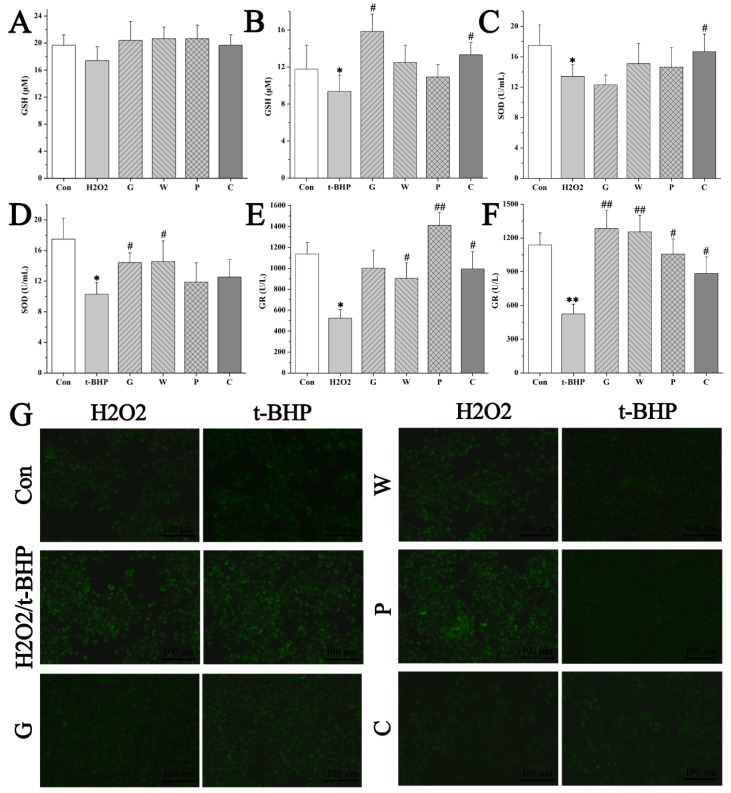
Efficacy evaluation of the crude and the processed Curcuma zedoaria on reactive oxygen species (ROS) related to oxidative damage by L02 assays**.** (**A**–**F**) The cellular glutathione (GSH), superoxide dismutase (SOD), and glutathione reductase (GR) levels of 55 μM H2O2 or 40 μM t-BHP model and/or the crude and the processed Curcuma zedoaria (83.33 mg/mL), respectively. (G) Morphological observation and relative fluorescence intensity of L02 cells induced by 55 μM H2O2 or 40 μM t-BHP and/or the crude and the processed Curcuma zedoaria (83.33 mg/mL) by ROS assay under the fluorescence microscope (100×). Con, control group treated with phosphate buffered saline (PBS); H2O2/t-BHP, model group treated with 55 μM H2O2 or 40 μM tertbutyl hydfroperoxide (t-BHP); G, W, P, C, cells treated with *Curcuma kwangsiensis* S. G. Lee et C. F. Liang, *Curcuma wenyujin* Y. H. Chen et C.Ling, *Curcuma phaeocaulis* Val., and vinegar-processed Curcuma zedoaria at doses of 83.33 mg/mL, respectively. The fluorescence reflected ROS levels. The most representative fields are shown. ANOVA with the post hoc test was used to calculate the significance of the differences, *, and ** represents *p* < 0.05, and *p* < 0.01 compared with the control group; #, and ## represent *p* < 0.05, and *p* < 0.01 compared with the model group, respectively. All experiments were performed three times, and the results are expressed as the mean ± S.D.

**Figure 4 molecules-24-02073-f004:**
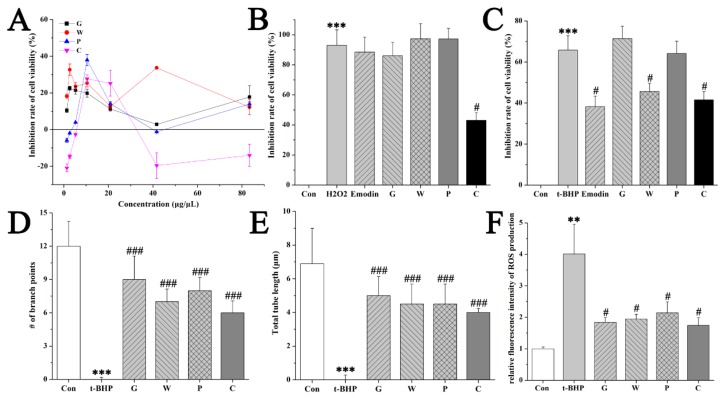
Efficacy evaluation of the crude and processed Curcuma zedoaria on oxidative liver injury by human brain microvascular endothelial cell (HBMEC) assays. (**A**) Inhibition rate of cell viability of HBMECs for MTT assays without 55 μM H2O2 or 40 μM t-BHP. (**B**–**C**) Inhibition rate of cell viability of HBMECs for MTT assays with 55 μM H2O2 or 40 μM t-BHP. (**D**–**E**) The crude and the processed Curcuma zedoaria confronted the inhibition of angiogenesis by 40 μM t-BHP. (**F**) Relative fluorescence intensity of HBMECs induced by 40 μM t-BHP and/or the crude and the processed Curcuma zedoaria (83.33 mg/mL) by ROS assay. Con, control group treated with PBS; Mod, model group treated with 55 μM H2O2 or 40 μM t-BHP; Pos, positive control group treated with emodin (12.5 μg/mL); G, W, P, C, cells treated with *Curcuma kwangsiensis* S. G. Lee et C. F. Liang, *Curcuma wenyujin* Y. H. Chen et C.Ling, *Curcuma phaeocaulis* Val., and vinegar-processed Curcuma zedoaria at doses of 83.33 mg/mL, respectively. ANOVA with the post hoc test was used to calculate the significance of the differences, *** represents *p* < 0.001 compared with the control group; #, and ### represent *p* < 0.05, and *p* < 0.001 compared with the model group, respectively. All experiments were performed three times, and the results are expressed as the mean ± S.D.

**Figure 5 molecules-24-02073-f005:**
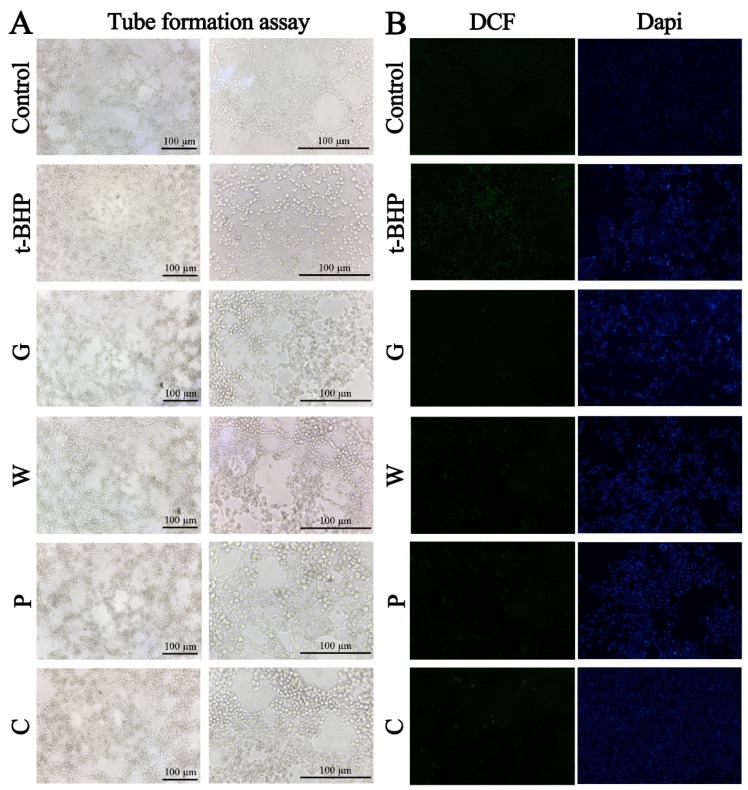
Efficacy evaluation of the crude and the processed Curcuma zedoaria on ROS related oxidative damage by HBMEC assays. (**A**) Angiogenesis of HBMECs under the microscope (100× and 200×). The most representative fields are shown. (**B**) Morphological observation of HBMECs induced by 40 μM t-BHP and/or the crude and the processed Curcuma zedoaria (83.33 mg/mL) by ROS and 4′,6-diamidino-2-phenylindole (DAPI) assay under the fluorescence microscope (50× and 100×). The most representative fields are shown. DCF, compounds with fluorescence for the reflection of ROS levels. Dapi, morphological observation using DAPI Staining. Con, control group treated with PBS; Mod, model group treated with 40 μM t-BHP; Pos, positive control group treated with emodin (12.5 μg/mL); G, W, P, C, cells treated with *Curcuma kwangsiensis* S. G. Lee et C. F. Liang, *Curcuma wenyujin* Y. H. Chen et C.Ling, *Curcuma phaeocaulis* Val., and vinegar-processed Curcuma zedoaria at doses of 83.33 mg/mL, respectively. All experiments were performed three times, and the results are expressed as the mean ± S.D.

**Figure 6 molecules-24-02073-f006:**
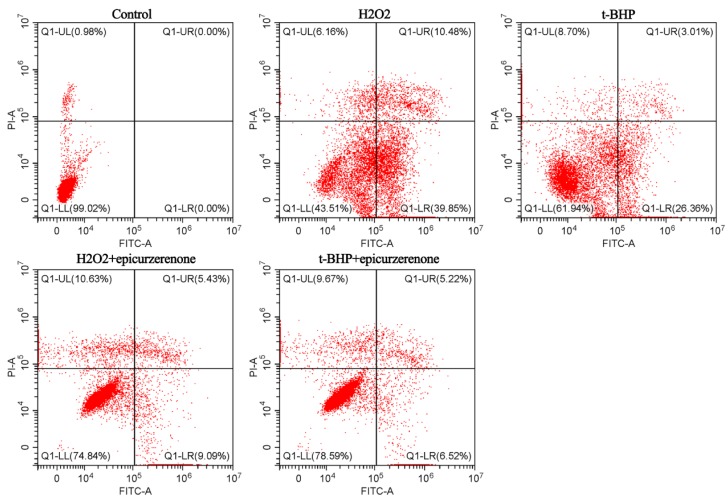
Epicurzerenone (32 μM) inhibited apoptosis of L02 cells induced by 55 μM H2O2 or 20 μM t-BHP using Annexin V-FITC/PI staining.

**Table 1 molecules-24-02073-t001:** Discriminatory components of the crude and the processed Curcuma zedoaria.

No.	RT	Compound Name	*p*.value	FDR	VIP	p(corr)	Fisher’s LSD
V7	7.96	4,5-di-2-furanyl-4,5-Octanediol	0.0000	0.0000	2.7395	0.3145	P - C; P - G; P - W
V10	19.12	6-Methyl-2-phenyl-7-(2,4-dimethylphenylmethyl)indolizine	0.0024	0.0108	0.3329	0.2820	G - C; G - W;
V13	27.20	beta-Myrcene	0.0046	0.0170	1.0252	−0.6188	W - C; W - G; W - P
V14	27.78	10-epi-gamma-Eudesmol	0.0000	0.0000	2.5892	0.1428	G - C; G - P; G - W
V25	35.58	Epicurzerenone	0.0000	0.0000	2.9176	0.5755	C - G; C - P; C - W; G - P; G - W
V35	42.36	(E,E)-3,7-dimethyl-10-(1-methylethylidene)-3,7-Cyclodecadien-1-one	0.0060	0.0204	1.3905	−0.5733	W - C; W - G; W - P

The significant differences were generated from the Student’s *t*-test or Mann-Whitney U test when the Student’s t-test was not suitable. G, W, P, C, the crude (G: *Curcuma kwangsiensis* S. G. Lee et C. F. Liang; W: *Curcuma wenyujin* Y. H. Chen et C.Ling; P: *Curcuma phaeocaulis* Val.) and vinegar-processed (C) Curcuma zedoaria. VIP: variable importance for projection.

**Table 2 molecules-24-02073-t002:** Quality marker identification and the integrated peak area in the crude and the processed Curcuma zedoaria.

No.	RT	Compound Name	Match Factor	CAS#	Formula	#group Integrated Peak Area in (the Mean ± S.D.)			
						C	G	P	W
V25	35.58	Epicurzerenone	94.96	20085-85-2	C15H18O2	3016994 ± 107555	577055 ± 150532	126842 ± 19496	44191 ± 4842

G, W, P, C, the crude (G: *Curcuma kwangsiensis* S. G. Lee et C. F. Liang; W: *Curcuma wenyujin* Y. H. Chen et C.Ling; P: *Curcuma phaeocaulis* Val.) and vinegar-processed (C) Curcuma zedoaria.

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
