# Peer review of "Identification of a Quality Marker of Vinegar-Processed Curcuma Zedoaria on Oxidative Liver Injury"

_molecules, 2019, doi:10.3390/molecules24112073_

Round 1

Reviewer 1 Report

Identification of a Quality marker of the vinegar processed curcuma zedoaria on oxidative liver injury

The introduction section looks to short. I suggest to increase it by including previous experimental work on the curcuma

The discussion section is too short. I suggest to increase it by focus on the main results of this study. Also I suggest to discuss other molecular mechanisms and pathways involved in the hepatoprotective activities of curcuma zedoaria

Some references are missing:

Akter J1, Hossain MA2, Takara K3, Islam MZ4, Hou DX5.

Antioxidant activity of different species and varieties of turmeric (Curcuma spp): Isolation of active compounds. Comp Biochem Physiol C Toxicol Pharmacol. 2019 Jan;215:9-17.

Lan TTP1,2, Huy ND3, Luong NN1, Nghi NV1, Tan TH3, Quan LV1, Loc NH1. Identification and Characterization of Genes in the Curcuminoid Pathway of Curcuma zedoaria Roscoe. Curr Pharm Biotechnol. 2018;19(10):839-846.

Abbreviations need to check along the text. Some of them are redundant

Author Response

We tried our best to improve the manuscript and made some changes in the manuscript. These changes will not influence the content and framework of the paper. Special thanks to you for your good comments.

Point 1: The introduction section looks to short. I suggest to increase it by including previous experimental work on the curcuma

Response 1: Thank you very much for your beneficial comments! Considering the Reviewer’s suggestion, the introduction section contained previous experimental work on Curcuma zedoaria has been added as one important part of Introduction as follows, along with related references (Introduction & References, Page 2).

Recent pharmacological studies have shown that Curcuma zedoaria has a wide range of pharmacological effects in anti-cancer, pharmacological effects on cardiovascular and cerebrovascular system, anti-fibrosis, anti-inflammatory and analgesic, anti-bacterial and anti-virus, hypoglycemia, anti-oxidation, anti-dysmenorrhea, etc [9]. Curcuma zedoaria has a wide range of pharmacological effects on rectal cancer, gastric cancer, liver cancer, lung cancer and cervical cancer and so on; on cardiovascular and cerebrovascular system, such as strong anti-platelet aggregation, anti-thrombosis, blood lipid regulation, anti-atherosclerosis and protective effects on ischemic cerebral apoplexy; as well, Curcuma zedoaria has anti-liver, kidney and pulmonary fibrosis effects [10, 11]. In recent years, it has been found that Curcuma zedoaria has obvious hypoglycemic and antioxidant effects with main active constituents including volatile oil, curcumin, curcumol, and beta-elemene and so on [12, 13]. 

9.         Jung, E.B., et al., Curcuzedoalide contributes to the cytotoxicity of Curcuma zedoaria rhizomes against human gastric cancer AGS cells through induction of apoptosis. J Ethnopharmacol, 2018. 213: p. 48-55.

10.       Hadisaputri, Y.E., et al., Molecular characterization of antitumor effects of the rhizome extract from Curcuma zedoaria on human esophageal carcinoma cells. Int J Oncol, 2015. 47(6): p. 2255-63.

11.       Mou, S., et al., Curcumin inhibits cell proliferation and promotes apoptosis of laryngeal cancer cells through Bcl-2 and PI3K/Akt, and by upregulating miR-15a. Oncol Lett, 2017. 14(4): p. 4937-4942.

12.       Tariq, S., et al., Phytopreventive antihypercholesterolmic and antilipidemic perspectives of zedoary (Curcuma Zedoaria Roscoe.) herbal tea. Lipids Health Dis, 2016. 15: p. 39.

13.       Akter, J., et al., Antioxidant activity of different species and varieties of turmeric (Curcuma spp): Isolation of active compounds. Comp Biochem Physiol C Toxicol Pharmacol, 2019. 215: p. 9-17.

Point 2: The discussion section is too short. I suggest to increase it by focus on the main results of this study. Also I suggest to discuss other molecular mechanisms and pathways involved in the hepatoprotective activities of curcuma zedoaria

Response 2: We have replenished this part according to the Reviewer’s suggestion. The details are shown as follows and have been revised in the manuscript (Discussion & References, Page 14-15).

Studies have confirmed that several main active ingredients of Curcuma zedoaria, such as curcumin, elemene and curcumol, have definite therapeutic effects on liver diseases [9, 24]. It has been found that curcumin has anti-hepatic injury effects, which are mainly manifested in scavenging free radicals, inhibiting inflammation and inhibiting the activation of hepatic stellate cells (HSC) via inhibiting oxidative damage induced by ROS and inflammatory response induced by NF-kappa B [25, 26]. Elemene can induce apoptosis of tumor cells, inhibit the growth of tumor cells and improve immune response of the body. Curcumol can inhibit hepatic fibrosis by inhibiting the expression of cytokines, induce apoptosis of tumor cells via inhibiting gene expression [27]. These literatures further confirm the therapeutic potential of Curcuma zedoaria on liver diseases, and its main mechanism is related to anti-oxidative stress and anti-inflammation [24, 27, 28]. This is consistent with our experimental results. Interestingly, few activity reports of epicurzerenone have been found based on the result of the searching database, which stimulated our interest in further study of this compound. 

9.         Jung, E.B., et al., Curcuzedoalide contributes to the cytotoxicity of Curcuma zedoaria rhizomes against human gastric cancer AGS cells through induction of apoptosis. J Ethnopharmacol, 2018. 213: p. 48-55.

24.       Lan, T.T.P., et al., Identification and Characterization of Genes in the Curcuminoid Pathway of Curcuma zedoaria Roscoe. Curr Pharm Biotechnol, 2018. 19(10): p. 839-846.

25.       Bao, W., et al., Curcumin alleviates ethanol-induced hepatocytes oxidative damage involving heme oxygenase-1 induction. J Ethnopharmacol, 2010. 128(2): p. 549-53.

26.       Yao, Q., et al., Curcumin ameliorates intrahepatic angiogenesis and capillarization of the sinusoids in carbon tetrachloride-induced rat liver fibrosis. Toxicol Lett, 2013. 222(1): p. 72-82.

27.       Chen, W., et al., Beta-elemene inhibits melanoma growth and metastasis via suppressing vascular endothelial growth factor-mediated angiogenesis. Cancer Chemother Pharmacol, 2011. 67(4): p. 799-808.

28.       Rajagopalan, R., S. Sridharana, and V.P. Menon, Hepatoprotective role of bis-demethoxy curcumin analog on the expression of matrix metalloproteinase induced by alcohol and polyunsaturated fatty acid in rats. Toxicol Mech Methods, 2010. 20(5): p. 252-9.

Point 3: Some references are missing:

Akter J1, Hossain MA2, Takara K3, Islam MZ4, Hou DX5. Antioxidant activity of different species and varieties of turmeric (Curcuma spp): Isolation of active compounds. Comp Biochem Physiol C Toxicol Pharmacol. 2019 Jan;215:9-17.

Lan TTP1,2, Huy ND3, Luong NN1, Nghi NV1, Tan TH3, Quan LV1, Loc NH1. Identification and Characterization of Genes in the Curcuminoid Pathway of Curcuma zedoaria Roscoe. Curr Pharm Biotechnol. 2018;19(10):839-846.

Response 3: Considering the Reviewer’s suggestion, we have replenished key references as follows, which has been added as Reference 13, 24 in the manuscript. As well, some related references have been added.

Point 4: Abbreviations need to check along the text. Some of them are redundant.

Response 4: Considering the Reviewer’s suggestion, we have revised this part in the manuscript as follows.

Abbreviations: TCM, traditional Chinese medicine; L02, normal hepatic cell line;  HBMEC, human brain microvascular endothelial cell; t-BHP, tertbutyl hydfroperoxide; total AOC, total antioxidant capacity.

We appreciate for Editors and Reviewers’ warm work earnestly, and hope that the correction will meet with approval. Once again, thank you very much for your comments and suggestions.

Hai-min Lei

May 16, 2019

Reviewer 2 Report

Dear Authors,

the manuscript has merit, however there are a few important points that need explanation:

Why epicurzerenone was chosen as quality marker? In my opinion, this compound should also be tested on liver cells as hepatoprotective agent, not only the extracts. You have mentioned that this compound is associated with this activity. On what basis?

What solvent of the extracts was used in tests with the cells? Did the solvent have influence on the cells? What was it final concentration?

Please check the legend of table (number 4?) and fig. 5 (H2O2? in the text and significant differences?).

Please check the description of subsection 2.3. There are a lot of mistakes (line 14-16) and the numbers of figures are muddled.

The statements are repeated in lines 15-17 on page 16.

Author Response

We tried our best to improve the manuscript and made some changes in the manuscript. These changes will not influence the content and framework of the paper. Special thanks to you for your good comments.

Point 1: Why epicurzerenone was chosen as quality marker? In my opinion, this compound should also be tested on liver cells as hepatoprotective agent, not only the extracts. You have mentioned that this compound is associated with this activity. On what basis?

Response 1: Thank you very much for your beneficial comments making our manuscript more convincing. Considering the Reviewer’s suggestion, the effect of epicurzerenone on apoptosis in L02 cells induced by H2O2/t-BHP by flow cytometric analysis has been replenished in the Quality marker identification section (Figure 6). Based on the results of discriminatory components analysis (Table 1-2) and the cell-based biological assay (Figure 6), epicurzerenone was identified as the Quality marker of vinegar-processed Curcuma zedoaria associated with the hepatoprotective activity.

Further, the effect of epicurzerenone on apoptosis in L02 cells induced by H2O2/t-BHP were determined by flow cytometric analysis. The cells were treated by 55 μM H2O2 or 20 μM t-BHP with/without epicurzerenone (32 μM) and then stained with both annexin Annexin V-FITC and PI. The flow cytometry observed four quadrant images: necrotic (Q1; Annexin−/PI+), late apoptotic (Q2; Annexin+/PI+), intact (Q3; Annexin−/PI−), and early apoptotic (Q4; Annexin+ /PI−) cells. The results were shown in Figure 6, the apoptosis ratios (including the early and late apoptosis ratios) increased to 50.33% (H2O2), 29.37% (t-BHP) while that of control and epicurzerenone groups were respectively 0.98% (control), 14.52% (H2O2+epicurzerenone), and 11.74% (t-BHP+epicurzerenone).

Figure 6. Epicurzerenone (32 μM) inhibits apoptosis of L02 cells induced by 55 μM H2O2 or 20 μM t-BHP using AnnexinV-FITC/PI staining.

Point 2: What solvent of the extracts was used in tests with the cells? Did the solvent have influence on the cells? What was it final concentration?

Response 2: 0.3 % DMSO (recognized nontoxic dose) in Dulbecco's modified eagle medium (DMEM) was used in tests with the cells. Considering the Reviewer’s suggestion, we have replenished “Materials and methods: Sample collection and preparation” section as follows (Page 15-16).

For cell-based biological assays, the crude and processed Curcuma zedoaria (833.33 mg/mL) were boiled with sterile double distilled water at 100 °C for 4 h based on the clinical dose and processing method and collected by freeze drying method. Then samples were dissolved in DMSO and diluted to final concentration (83.33 mg/mL), in which the content of DMSO was controlled in 0.3 % (recognized nontoxic dose).

Point 3: Please check the legend of table (number 4?) and fig. 5 (H2O2? in the text and significant differences?).

Response 3: Considering the Reviewer’s suggestion, we have revised this part in the manuscript.

Point 4: Please check the description of subsection 2.3. There are a lot of mistakes (line 14-16) and the numbers of figures are muddled.

Response 4: Thank you very much for your beneficial comments. We have revised this part in the manuscript as follows considering the Reviewer’s suggestion.

Point 5: The statements are repeated in lines 15-17 on page 16.

Response 5: Considering the Reviewer’s suggestion, we have revised this part in the manuscript.

We appreciate for Editors/Reviewers’ warm work earnestly, and hope that the correction will meet with approval. Once again, thank you very much for your comments and suggestions.

Hai-min Lei

May 16, 2019

Round 2

Reviewer 2 Report

Dear Authors,

some of the suggestions have not been corrected: please check the subsection 2.3 (page 9) and check the references to the figures (in this subsection should not be Figure 4A instead 3A? (line 12), in line 20 there is reference to figure 3, why? similarly in lines: 23 and 27.

In line 25 - it should not be Figure 5B? please check.

Next, the statements in lines 4-6 on page 18 have not been corrected.

Author Response

Special thanks to you for your further comments. We tried our best to improve the manuscript and made some changes in the manuscript. These changes will not influence the content and framework of the paper.

Point 1: Some of the suggestions have not been corrected: please check the subsection 2.3 (page 9) and check the references to the figures (in this subsection should not be Figure 4A instead 3A? (line 12), in line 20 there is reference to figure 3, why? Similarly in lines: 23 and 27.

Response 1: Thank you very much for your beneficial comments. Considering the Reviewer’s suggestion, we have revised this part in the manuscript as follows and checked our manuscript carefully.

2.3. Comparison of hepatoprotective activities by HBMEC assays

The vascular endothelial cells has an inductive effect on liver development before the establishment of a blood flow [20]. Endothelial cells are not only the target cells of free radical, but also the important tissue of free radical production [21]. Stimulating endothelial cells angiogenesis could form the basis of a therapeutic scheme aimed toward protection of hepatocytes, which may have therapeutic potential for preservation of organ function in certain liver disorders [20]. Thus, the efficacy of the crude and processed Curcuma zedoaria on oxidative liver injury was subsequently evaluated on HBMEC assays including proliferation, angiogenesis, apoptosis and ROS-production. The cell viability of several concentrations of the crude and processed Curcuma zedoaria on HBMECs was firstly evaluated by MTT assay, and the result showed that the dose in this manuscript was reliable because of the under 50 % inhibition of cell viability (Figure 4A). Then, 55 μM H2O2 or 40 μM t-BHP was chosen to be the model conditions according to the preliminary studies. MTT assay revealed the inhibiting effect of H2O2 or t-BHP on the viability of HBMECs by contrast of control group. And this damage was protected by the crude (G, W, P group) and processed (C group) Curcuma zedoaria (83.33 mg/mL) treated with a significant difference (p < 0.05) (Figure 4B-C). Besides, subsequent studies only applied with t-BHP model on account of the strong damage of 55 μM H2O2. Further, t-BHP resulted in the damage of HBMECs within 4 h with the blockage of angiogenesis, the increase of apoptosis cells and ROS production with a significant difference (p < 0.05) (Figure 4-5). In contrast to the model group, the processed Curcuma zedoaria-treated HBMECs survived a damaged dose of toxicants. Specifically, the crude and processed Curcuma zedoaria could confront the blockage of angiogenesis with a significant difference (p < 0.05) (Figure 4D-E, 5A). DAPI staining was used for apoptosis detection of cells by morphological analysis. The nucleus of HBMECs in control group were complete and oval, with weak fluorescence intensity (Figure 5B). The relative fluorescence intensity was inhibited by the crude and processed Curcuma zedoaria, showing the potent effect on suppressing ROS production (Figure 4E, 5B). These results indicated that the crude and processed Curcuma zedoaria could reverse the oxidative damage caused by H2O2/t-BHP. However, damaged HBMECs were recuperated by the processed Curcuma zedoaria far than the crude ones.

Point 2: In line 25 - it should not be Figure 5B? please check.

Response 2: Considering the Reviewer’s suggestion, we have revised this part in the manuscript as above and checked our manuscript carefully.

Point 3: Next, the statements in lines 4-6 on page 18 have not been corrected.

Response 3: Considering the Reviewer’s suggestion, we have revised this part in the manuscript and checked our manuscript carefully.

4.12. Statistical analysis

The data were analyzed with the SPSS software program (version 22.0, Chicago, IL, USA). One-way analysis of variance (ANOVA) with the post hoc test followed by Student’s t-test (the Mann-Whitney U test was used when the t-test was not suitable) was used for the evaluation of significant differences of the results. The differences were considered to be statistically significant when p < 0.05 and highly significant when p < 0.01. FDR correction was not used during the univariate analysis of the metabolomics analysis because the metabolites with small p-values had been examined by building the classification model [31].

We appreciate for Editors/Reviewers’ warm work earnestly, and hope that the correction will meet with approval. Once again, thank you very much for your comments and suggestions.

Hai-min Lei

May 22, 2019